# Influence of palliative care policy on place of death for people with different cancer types: a nationwide' register study

**Joakim Öhlén**[1,2,3], **Nyblom Stina**[2,3,4], **Ozanne Anneli**[1,5], **Nilsson Stefan**[1,2,6], **Gyllensten Hanna**[1,2], **O'Sullivan Anna**[7,8], **Fürst Carl Johan**[9], **Larsdotter Cecilia**[8]

1 Institute of Health and Care Sciences, Sahlgrenska Academy, University of Gothenburg, Gothenburg, Sweden, 2 Centre for Person-Centred Care (GPCC), University of Gothenburg, Gothenburg, Sweden, 3 Palliative Centre, Sahlgrenska University Hospital Västra Götaland Region, Gothenburg, Sweden, 4 Institute of Medicine, Sahlgrenska Academy, University of Gothenburg, Gothenburg, Sweden, 5 Department of Neurology, Sahlgrenska University Hospital, Gothenburg, Sweden, 6 Queen Silvia Children´s hospital, Sahlgrenska University Hospital, Gothenburg, Sweden, 7 Department of Health Care Sciences, Marie Cederschiöld University, Stockholm, Sweden, 8 Department of Nursing Science, Sophiahemmet University, Stockholm, Sweden, 9 The Institute for Palliative Care, Faculty of Medicine, Lund University, Lund, Sweden

* joakim.ohlen@gu.se

**Citation:** Öhlén J, Stina N, Anneli O, Stefan N, Hanna G, Anna O, et al. (2025) Influence of palliative care policy on place of death for people with different cancer types: a nationwide' register study. PLoS ONE 20(3): e0320086. https://doi.org/10.1371/journal.pone.0320086

## Abstract

This study investigated the impact of a national palliative care policy introduced in 2013. The hypothesis was that end-of-life and palliative care policy shape healthcare services, which in turn influence service utilisation and ultimately place of death for people dying from cancer. The aim was to identify longitudinal trends in place of death for people with cancer in Sweden. A population-level longitudinal trend in place of death study was performed, based on register data of all adults aged 18 or above with a cancer diagnosis as underlying cause of death in Sweden between 2013-2019. Data were retrieved from registers at the Swedish National Board of Health and Welfare and Statistics Sweden. In addition to a more descriptive overview of place of death (hospital, home, nursing home, and other places), linear regression models were used to analyse trends in place of death and associated socio-demographic factors, and healthcare services and utilisation. Dying in hospital was most common (48.7%), followed by nursing home (25.6%) and own home (23.5%), and differed according to sex, age, marital status, type of cancer, healthcare regions, and utilising specialised palliative care services at death or not. From 2013 to 2019 the total number of home deaths increased from 21.8% to 24.7%, whereas hospital deaths decreased from 49.2% to 47.1%. For people residing in their own home, there was a downward trend for dying in hospital, while no trend was detected for people residing in nursing homes. The identified trend had cross-regional variations and inconsistencies. In conclusion, the hypothesis was not confirmed. The results point to inequity in palliative cancer care, and need for national governance strategies and improved integration of palliative care in national healthcare structures.

**Data availability statement:** The data for this study is available from each register holder. Certain restrictions may apply: The National Board of Health and Welfare (https://bestalla-data.socialstyrelsen.se/data-for-forskning/), the Statistics Sweden (https://www.scb.se/en/services/ordering-data-and-statistics/), and the Swedish Register for Palliative Care (https://palliativregistret.se/forskare/datauttag/). Programming codes are available from the authors upon reasonable request.

**Funding:** The study was supported by the Swedish state under the agreement between the Swedish government and the county councils the ALF-agreement (ALFGBG-965941); and The Swedish Cancer Society (grant no. 21 1580Pj01H). "The funders had no role in study design, data collection and analysis, decision to publish, or preparation of the manuscript.

**Competing interests:** The authors have declared that no competing interests exist.

# INTRODUCTION

Death caused by cancer accounts for almost two million deaths annually in Europe, with lung, colorectal, breast, and pancreatic cancer being the most common underlying causes of death [1]. Hence, cancer care at end of life is a fundamental issue for decision-making at a population level. Place of death is considered of importance for healthcare decision-making and policy, as it may reflect prerequisites for palliative care in populations and thus indicate that the organisation and provision of palliative care [2,3] impacts quality of life for people who are severely ill and dying of cancer [4].

The desired overall outcome across discourses and populations is that death occurs in the patient's preferred place, which is commonly the patient's own home [5–7]. For this reason, home death is generally used as a proxy measure for fulfilment of the patient's preferred place of death [8]. Home deaths are associated with the patient's and family members' preferences, and outcomes in terms of wellbeing, peace and grief are generally better for those dying at home than in hospital [9,10].

For a number of countries, the proportion of home deaths for people with cancer is higher than home deaths due to non-cancer causes [11], including in Sweden, which is the focus of this study. For people dying of cancer in Europe, the proportion of hospital deaths is reported to vary from 18%-76% [11,12] and in Sweden it is 51% [13]. Potential overtreatment is reported for one-third of people aged 65 and older dying of cancer, which may impact place of death [14]. Moreover, both increasing and decreasing trends for home deaths are reported, with increasing home deaths and decreasing hospital deaths generally being regarded as progress [15]. However, hospital, as the most common actual place of death, does not reflect people's preferences for place of death.

Hospital deaths are found to be associated with living in areas of high deprivation, being in work [16] and being single, widowed or divorced, of older age, and having haematological or lung cancer [17]. Home deaths are found to be associated with high income and living in rural areas [16]. Nevertheless, there are contrary results reported for living in rural areas [13]) and these relate to the patients' own preferences, and whether living with family member(s), and also reduced functional status [17] and lack of healthcare resources [11]. In fact, a Cochrane review found that patients receiving specialised palliative care are more likely to die at home than others [8]. Further, home death or death in a residential nursing home, as well as enrolment in a specialised palliative service and having official palliative care status, are considered indicators of appropriate quality of care at the end-of-life for people with cancer, while emergency visits and hospital admissions are regarded as indicators of inappropriate care, i.e., futile treatment and insufficient symptom relief [18].

For most people in the population dying of cancer, to facilitate their proper care and preferred place of death (and in particular, for this to be at home) it is imperative that a national policy is in place for equitable palliative care at the end-of-life. Several European countries are found to have palliative care integrated in their national healthcare structures [19,20]. However, evaluations of such integration in European countries also reveal a different view, whereby different reports find varying degrees of integration in some countries, including Sweden [21]. Hence, national integration of palliative care may take different forms. Palliative care as a special orientation in end-of-life care was initially introduced based on practice examples and a specific philosophical approach to care, initially for the care of dying people [3]. Today, however, there is research-based evidence – not yet implemented – to introduce and integrate palliative care with oncological treatment and care, and this may include patient referral to specialised palliative care services [22,23].

Based on previous research and theory [24], the hypothesis is that end-of-life and palliative care policy shape healthcare service types, locations and practices, which in turn influence

service utilization and ultimately place of death. These factors may also be influenced by patients' diagnoses and socio-demographics. Enrolment in a specialised palliative service may increase the prevalence of home death [8,25]. In the decades to come, the provision of care at the end-of-life for populations with cancer will become an increasingly critical issue including in Sweden, given the expected increase of older people including cancer as a major cause of death. Consequently, knowledge about longitudinal trends in place of death due to cancer is of increasing importance for overall healthcare decision-making. Since Sweden initiated a national policy for palliative care in 2012 [26,27], there is a case for investigating its impact on prerequisites for palliative care. This is also motivated by the results of a recent study on place of death for the total population, which indicates a higher proportion of deaths in hospitals for cancer patients than for other diagnostic groups [28].

With an interest in the development of palliative cancer care, the aim of this study was to identify longitudinal trends in place of death on a population level for people with cancer in Sweden. The research questions were: What is the distribution of place of death and the healthcare service utilisation in specialised palliative care services at the end-of-life in the Swedish cancer population? Are there longitudinal trends concerning place of death between 2013-2019? Are there associations between place of death and socio-demographics, healthcare services and healthcare utilisation?

## Materials and Methods

### Design

The study design was longitudinal and examined nation-wide population-level trends in place of death based on registry data [26] of all adults 18 or older with a cancer diagnosis as underlying cause of death and a registered place of death (based on death certificates) in Sweden between 2013–2019. The chosen time period was informed by the hypothesis that the first national palliative care policy [27,28] in Sweden was implemented in 2013, with clinical guidelines developed by and targeting health professionals at the point of care [27] and knowledge support developed at governmental level primarily for governance [28]. The end year was due to the assumption that the Covid-19 pandemic influenced place of death. Data for the period was retrieved from the National Board of Health and Welfare in Sweden (NBHW), Statistics Sweden, and the Swedish Register for Palliative Care (SRPC), see Table 1. Data were linked between registers on the individual level through the standard, state-issued personal identity numbers, and then anonymised (replacing the personal identity numbers with unique serial numbers) before being accessed by the researchers.

The study was ethically assessed by the Swedish Ethical Review Authority and the committee had no objection to the proposed research project (no. 2019-05213, 2020-01758). The Ethical Review Authority gave approval for the study to be conducted without ethical vetting because all study participants were deceased and information about deceased persons is not considered personal data. Thus, no intervention could be performed on any one research person and there would be no treatment of personal data. None of the authors had access to any information that could identify individual participants in the data set.

### Setting

Sweden, with a population of ten million, is characterized by increasing urbanisation and an increasing proportion of older people. It has three metropolitan areas, 50 municipalities with more than 50,000 inhabitants, and larger, sparsely populated areas. The incidence of cancer increases in proportion to the age of the population. The overall preferred place of death is the own home, although robust studies thereof are lacking [7].

**Table 1. Type of data, variables and categorisation as related to data sources.**

| Type of data | Variables and categorisation | Data sources (register holders) |
|---|---|---|
| Outcome | Place of death<br>Home (privately owned or rented)<br>Hospital (unspecified speciality; healthcare provided by regions)<br>Nursing home (i.e., including residential care settings and other forms of group dwellings)<br>Other (e.g., public places, roads, workplace) | Death certificate register (NHWB[a]) |
| Medical diagnoses (underlying cause of death) | Types of cancer, based on ICD-10 codes [30] [b]<br>Lower gastrointestinal (C18-C21)<br>Upper gastrointestinal (C15-C17, C220-C224, C227, C229, C23, C240-C241, C248-C254, C258-C261, C268-C269)<br>Pulmonary (C34)<br>Breast & gynaecological (C50-C58)<br>Prostate and urinary tract (C61, C64-C66, C670-C671, C674-C680, C688-C689)<br>Haematological (C81-C91, C920-C925, C927C931, C933, C939-C940, C942-C944, C946-C947, C950-C951, C959-C960. C962, C966-C969)<br>Malignant melanoma and sarcoma (C431-C439, C499)<br>Other (all other C-codes) | Death certificate register (NHWB[a]) |
| Official palliative care status | Palliative care diagnosis[c]<br>ICD-code Z51.5: yes/no | Death certificate register (NHWB[a]) |
| Socio-demographics | Sex<br>Male/ Female<br>Age<br>Divided into 10-year categories<br>Marital status<br>Unmarried/ Married/ Widowed/ Divorced<br>Educational attainment<br>No formal or elementary education/ Lower secondary education/ Higher secondary education/ Higher education<br>Birth country<br>Sweden/ Other countries<br>Household situation<br>Living in single-person household/ Living in multi-person household<br>Children under 18 years in the household<br>Yes/ No<br>Living situation<br>Home; own residence/ Home; rented residence/ Nursing home/ Other | Death certificate register (NHWB[a])<br>National socioeconomic registers (StatS[d]) |
| Health service characteristics | National healthcare regions<br>North region/ Uppsala-Örebro region/ Stockholm region/ West region/ Southeast region/ South region<br>Residing in urban area<br>Yes/ No | Regional organisation of cancer care and treatment [e]<br>National socioeconomic registers (StatS[d]) |
| Healthcare service utilisation at the end of life | Hospital transfers during last month of life<br>One transfer/ Two or more transfers<br>Emergency care during last month of life<br>One visit/ Two or more visits<br>Specialised palliative care service at death[f]<br>No/ Yes | Patient data register (NHWB[a])<br>Swedish Register for Palliative Care |

Notes.

[a]National Board of Health and Welfare.

[b]Individual ICD-10 codes listed below.

[c]Palliative care is a medical classification listed by WHO 'to indicate a reason for care' within the category Persons encountering health health services for specific procedures and health care [30].

[d]Statistics Sweden.

[e]Well established nation-wide.

[f]As related to place of death, the types of specialised palliative care services were specialised palliative home care (home), specialised palliative hospital care (hospital), and municipality hospice care (nursing home).

The tax-financed Beveridge oriented national healthcare system has national governance, with overall responsibility at the Ministry of Health and Social Affairs and the government agencies. This is combined with highly decentralised governance for regions and municipalities allocating resources, monitoring and accountability [29]. By law, patient autonomy is primary. To enable patient-focused, accessible and equitable cancer care, six regional cancer centres coordinate cancer care. The responsibility for healthcare is decentralised and divided, with the regions being responsible for hospitals and primary care, and the municipalities for home care and nursing homes. Nursing homes are designed to provide service and care (with 24/7 staffing of assistant nurses and on demand access to registered nurses and physicians) while also supporting autonomy and taking the form of residential care housing. Physicians from primary care services are collaboratively integrated in both nursing home and home care services to secure healthcare, including care at the end-of-life and dying. In the study period, the number of hospital beds per 1,000 people decreased from 2.6 to 2.1, while the health and social employment density per 1,000 people (head counts) increased from 79 to 83.

Palliative care practice development and research has evolved since the 1970s and accelerated since the beginning of the 2000s. Practice development has assumed different forms, with emphasis on specialised palliative care services in some healthcare regions and an emphasis on general, non-specialised, palliative care in others.

## Study variables

The primary outcome for all analyses of place of death was categorised into four distinct alternatives: hospital, home, nursing home, and other places. Individuals utilising specialised palliative care services or not at death were distinguished (not applicable for other places). Cancer diagnoses registered according to ICD-10 [30] were grouped into cancer categories. The specific diagnostic code for palliative care, ICD-10 51.5 as reason for care [30], was also included as an indicator of official palliative care status (Table 1).

Additional factors with known impact on place of death were included in the analyses [24]. Socio-demographics consisted of sex, age, marital status, educational attainment, birth country, household, having children under the age of 18, and living situation. Healthcare services data consisted of healthcare region and whether or not residing in an urban area, to reflect proximity to healthcare services. Healthcare service utilization data consisted of hospital transfers and emergency care during the last month of life, and specialised palliative service at death or not (for values and specification, see Table 1). The variable 'specialised palliative service at death' was categorised into specialised palliative home care (home), specialised palliative hospital care (hospital), and municipality hospice care (specialised nursing home). Accordingly, non-specialised palliative care services were provided in general home care services, hospital wards in various medical specialities (within regions), and nursing homes (within municipalities).

## Analyses

Descriptive statistics were calculated for distribution in the four overall places of death and other variables, and as related to the six healthcare regions, cancer types and utilisation of specialised palliative care services or not at death.

Linear regression modelling was performed to investigate longitudinal trends in place of death and associated factors. Although the outcome was not continuous, the linear model showed an excellent fit in explaining variations in place of death at the population level. Robust standard errors (HC3) were employed to account for violations of the assumption of

normality. Two separate analyses were performed: a) place of death in hospital as dependent variable for all individuals residing in their own home (and dying in either hospitals or at home), and b) place of death in hospital as dependent variable for individuals older than 60 residing in nursing homes (and dying in either hospitals or nursing homes). 'Other places' were excluded due to small numbers. Results are presented as percentage points change per year of people dying in hospital with 95% confidence intervals (CIs). The coefficient of determination ($R^2$) was used to summarise the strength of association at the population (aggregate) level and presented in the figures. This was calculated using linear regression on observed relative frequencies vs time. Interaction analyses were performed to evaluate for influence of covariates on longitudinal trends in place of death.

Due to the large sample size, all significance tests were two sided and conducted at the α=0.001 significance level. Statistical analyses were performed using SAS/STAT Software, version 9.4 of the SAS System for Windows (SAS Institute Inc., Cary, NC).

## RESULTS

### Distribution of place of death

Of the 152,462 adults (47.8% women) who died in Sweden from 2013 to 2019 with cancer as an underlying cause of death, 48.7% died in hospital, 25.6% in a nursing home, 23.5% at home and 2.3% in other places. Overall, the percentages of women/men who died in hospital, nursing home and at home were 48.4%/49.0%, 27.2%/24.1% and 22.0%/24.9% respectively (Table 2; Cross regional distribution displayed in S1 Table). Of the total study population, 30.4% had a palliative care diagnosis, whereas 36.3% died in a specialised palliative care setting. The largest groups were married (45.7%), lived in multiple-person households (59.3%), had higher secondary school as their highest educational attainment (41.6%), resided in an urban area (86.5%), and, during the last month of life, had one or more hospital transfers (69.4%) and had no unplanned emergency visits (64.3%). The distribution of place of death and other variables for the total study population is presented in Table 2.

Healthcare service utilisation in specialised palliative care services at death varied by healthcare regions (Table 3). In the six different healthcare regions, the percentage of hospitals deaths utilising specialised palliative care services ranged from 8.7–41.9% (North Region lowest in five cancer types and Stockholm Region highest in all cancer types). Haematological cancer had the highest proportion of hospital deaths (61.6%), which varied regionally: the group with hospital deaths utilising specialised palliative care services at death ranged from 4.2–30.1%. The cancer type with the lowest proportion of hospital deaths was prostate and urinary tract cancer (40.4%), which within the healthcare regions varied from 6.9–37.3% for those with hospital deaths utilising specialised palliative care services. In nursing homes, prostate and urinary tract cancer (33.4%) was most frequent and upper gastrointestinal cancer (20.9%) the least frequent cancer type. At home, lower gastrointestinal cancer (28.2%) was most frequent and haematological cancer (16.2%) the least frequent cancer type (Table 2). See S2 Table for the overall cross-regional place of death distribution by cancer type, as related to utilising specialised palliative care services at death.

### Longitudinal trends in place of death

During the study period, the total number of home deaths increased from 21.8 to 24.7%, whereas the total number of hospital deaths decreased from 49.2 to 47.1%. Within the six healthcare regions, the largest proportional regional increase in home deaths was in the Southeast Region (5.3%) followed by the South Region (4.6%) and Uppsala-Örebro Region (4.3%), whereas no change was observed in the Stockholm Region. For nursing home deaths,

**Table 2. Distribution of place of cancer deaths by sex, age, cancer types and other variables in the total population from 2013 to 2019.**

| Variables[a] | Total (n = 152462) | Hospital (n = 74206) | Nursing home (n = 38986) | Home (n = 35784) | Other or unknown (n = 3486) |
|---|---|---|---|---|---|
| **Deaths by year** | | | | | |
| 2013 | 21,209 | 10,426 (49.2%) | 5,757 (27.1%) | 4,631 (21.8%) | 395 (1.9%) |
| 2014 | 21,448 | 10,549 (49.2%) | 5,704 (26.6%) | 4,853 (22.6%) | 342 (1.6%) |
| 2015 | 21,691 | 10,443 (48.1%) | 5,828 (26.9%) | 5,010 (23.1%) | 410 (1.9%) |
| 2016 | 21,857 | 10,854 (49.7%) | 5,190 (23.7%) | 5,276 (24.1%) | 537 (2.5%) |
| 2017 | 22,379 | 10,992 (49.1%) | 5,497 (24.6%) | 5,360 (24.0%) | 530 (2.4%) |
| 2018 | 21,916 | 10,590 (48.3%) | 5,522 (25.2%) | 5,232 (23.9%) | 572 (2.6%) |
| 2019 | 21,962 | 10,352 (47.1%) | 5,488 (25.0%) | 5,422 (24.7%) | 700 (3.2%) |
| **Sex** | | | | | |
| Male | 79,530 | 38,939 (49.0%) | 19,139 (24.1%) | 19,764 (24.9%) | 1,688 (2.1%) |
| Female | 72,932 | 35,267 (48.4%) | 19,847 (27.2%) | 16,020 (22.0%) | 1,798 (2.5%) |
| **Age at death, years** | | | | | |
| 18-29 | 424 | 245 (57.8%) | 33 (7.8%) | 136 (32.1%) | 10 (2.4%) |
| 30-39 | 986 | 592 (60.0%) | 82 (8.3%) | 278 (28.2%) | 34 (3.4%) |
| 40-49 | 3,313 | 1,939 (58.5%) | 336 (10.1%) | 910 (27.5%) | 128 (3.9%) |
| 50-59 | 10,013 | 5,774 (57.7%) | 1,179 (11.8%) | 2,764 (27.6%) | 296 (3.0%) |
| 60-69 | 28,643 | 16,268 (56.8%) | 4,163 (14.5%) | 7,372 (25.7%) | 840 (2.9%) |
| 70-79 | 49,659 | 25,963 (52.3%) | 10,262 (20.7%) | 12,200 (24.6%) | 1,234 (2.5%) |
| 80-89 | 45,237 | 19,046 (42.1%) | 15,708 (34.7%) | 9,709 (21.5%) | 774 (1.7%) |
| 90 + | 14,184 | 4,377 (30.9%) | 7,223 (50.9%) | 2,414 (17.0%) | 170 (1.2%) |
| **Cancer type; underlying cause of death** | | | | | |
| Lower gastrointestinal | 18,709 | 7,915 (42.3%) | 5,061 (27.1%) | 5,277 (28.2%) | 456 (2.4%) |
| Upper gastrointestinal | 29,689 | 14,572 (49.1%) | 6,208 (20.9%) | 8,156 (27.5%) | 753 (2.5%) |
| Pulmonary | 24,767 | 13,842 (55.9%) | 5,201 (21.0%) | 5,116 (20.7%) | 608 (2.5%) |
| Breast & gynaecological | 17,926 | 8,335 (46.5%) | 5,122 (28.6%) | 4,003 (22.3%) | 466 (2.6%) |
| Prostate and urinary tract | 24,934 | 10,082 (40.4%) | 8,325 (33.4%) | 6,054 (24.3%) | 473 (1.9%) |
| Haematological | 13,238 | 8,148 (61.6%) | 2,774 (21.0%) | 2,140 (16.2%) | 176 (1.3%) |
| Malignant melanoma and sarcoma | 4,230 | 2,090 (49.4%) | 1,035 (24.5%) | 998 (23.6%) | 107 (2.5%) |
| Other | 18,969 | 9,222 (48.6%) | 5,260 (27.7%) | 4,040 (21.3%) | 447 (2.4%) |
| **Palliative care diagnosis** | | | | | |
| No | 106,167 | 50,208 (47.3%) | 30,308 (28.5%) | 24,043 (22.6%) | 1,608 (1.5%) |
| Yes | 46,295 | 23,998 (51.8%) | 8,678 (18.7%) | 11,741 (25.4%) | 1,878 (4.1%) |
| **Marital status** | | | | | |
| Married | 69,615 | 35,386 (50.8%) | 12,365 (17.8%) | 20,324 (29.2%) | 1,540 (2.2%) |
| Unmarried | 19,480 | 10,136 (52.0%) | 4,997 (25.7%) | 3,812 (19.6%) | 535 (2.7%) |
| Widowed | 36,492 | 14,785 (40.5%) | 14,555 (39.9%) | 6,459 (17.7%) | 693 (1.9%) |
| Divorced | 26,827 | 13,865 (51.7%) | 7,065 (26.3%) | 5,182 (19.3%) | 715 (2.7%) |
| **Educational attainment** | | | | | |
| Higher secondary education | 63,380 | 32,318 (51.0%) | 14,472 (22.8%) | 14,974 (23.6%) | 1,616 (2.5%) |
| No formal or elementary education | 46,560 | 19,889 (42.7%) | 15,619 (33.5%) | 10,283 (22.1%) | 769 (1.7%) |
| Lower secondary education | 13,861 | 7,147 (51.6%) | 3,084 (22.2%) | 3,242 (23.4%) | 388 (2.8%) |
| Higher education | 26,323 | 13,770 (52.3%) | 5,211 (19.8%) | 6,677 (25.4%) | 665 (2.5%) |
| **Country of birth** | | | | | |
| Born in Sweden | 132,578 | 63,703 (48.0%) | 35,054 (26.4%) | 30,872 (23.3%) | 2,949 (2.2%) |
| Born outside Sweden | 19,871 | 10,495 (52.8%) | 3,931 (19.8%) | 4,909 (24.7%) | 536 (2.7%) |

*(Continued)*

**Table 2.** (Continued)

| Variables[a] | Total (n = 152462) | Hospital (n = 74206) | Nursing home (n = 38986) | Home (n = 35784) | Other or unknown (n = 3486) |
|---|---|---|---|---|---|
| Household situation | | | | | |
| Single-person household | 61,584 | 27,714 (45.0%) | 21,706 (35.2%) | 10,769 (17.5%) | 1,395 (2.3%) |
| Multi-person household | 90,372 | 46,200 (51.1%) | 17,149 (19.0%) | 24,947 (27.6%) | 2,076 (2.3%) |
| Children under 18 in the household | 6,572 | 3,574 (54.4%) | 903 (13.7%) | 1,870 (28.5%) | 225 (3.4%) |
| Living situation | | | | | |
| Home | | | | | |
| Owned residence | 96,866 | 49,061 (50.6%) | 20,212 (20.9%) | 25,290 (26.1%) | 2,303 (2.4%) |
| Rented residence | 40,903 | 20,616 (50.4%) | 10,404 (25.4%) | 8,869 (21.7%) | 1,014 (2.5%) |
| Nursing home | 7,859 | 1,576 (20.1%) | 5,774 (73.5%) | 467 (5.9%) | 42 (0.5%) |
| Other | 2,625 | 1,241 (47.3%) | 811 (30.9%) | 528 (20.1%) | 45 (1.7%) |
| Residing in urban area | | | | | |
| NO | 20,565 | 9,844 (47.9%) | 4,377 (21.3%) | 5,988 (29.1%) | 356 (1.7%) |
| Residing in urban area | 131,849 | 64,328 (48.8%) | 34,605 (26.2%) | 29,789 (22.6%) | 3,127 (2.4%) |
| Number of hospital transfers during last month of life | | | | | |
| None | 46,643 | 2,592 (5.6%) | 21,678 (46.5%) | 21,830 (46.8%) | 543 (1.2%) |
| One transfer | 59,216 | 35,145 (59.4%) | 12,271 (20.7%) | 10,252 (17.3%) | 1,548 (2.6%) |
| Two or more transfers | 46,603 | 36,469 (78.3%) | 5,037 (10.8%) | 3,702 (7.9%) | 1,395 (3.0%) |
| No of emergency department visits during last month of life | | | | | |
| None | 98,000 | 39,036 (39.8%) | 29,823 (30.4%) | 26,654 (27.2%) | 2,487 (2.5%) |
| One unplanned health care visit | 39,074 | 25,283 (64.7%) | 6,675 (17.1%) | 6,309 (16.1%) | 807 (2.1%) |
| Two or more unplanned health care visits | 15,388 | 9,887 (64.3%) | 2,488 (16.2%) | 2,821 (18.3%) | 192 (1.2%) |

Notes For categorical variables n (row %) is presented.

[a]Missing data: Age at death <0.00%, Marital status 0.03%, Educational attainment 1.5%, Country of birth <0.00%, Household situation 0.3%, Living situation 2.8%, Residing in urban area 0,03%; no missing data for the remaining variable

the largest decrease was in the South Region (3.7%) followed by Uppsala-Örebro Region (3.0%), and an increase in the Stockholm Region (1.6%) (Fig 1; S3 Table).

In the period 2013–2019 for the population residing in their own home (n = 103,836), there was an overall downward trend for dying in hospital (-0.51, CI: -0.65, -0.02)(Fig 2a). In addition to variations between healthcare regions (Fig 2a), interaction analyses showed significant differences in trends depending on palliative care diagnosis, marital status, and specialised care service (Fig 3b–3d). Both women and men showed a similar downward trend as the overall trend (Fig 3a). There was a stronger downward trend over time for dying in hospital for the age groups 80-89 years (-0.71, CI: -1.00, -0.43) and 90+ (-1.26, CI: -1.88, -0.64) than the overall trend (-0.51, CI: -0.65,-0.37), while there was no trend for the other age groups (Fig 3e). Within the cancer types prostate and urinary tract, haematological, and other cancers the downward trend for dying in hospital was stronger than the same trend for the overall cancer population (Fig 3c). For patients with a palliative care diagnosis and for utilisation of specialised palliative services, there was an upward trend over time for dying in hospital, while those without the diagnosis and not utilising specialised palliative care services contributed to a downward trend for dying in hospital (Fig 3b,3c). Widowed and divorced people showed a greater downward trend for dying in hospital than the overall trend (Fig 3d). In addition, there were large variations in levels of hospital deaths as related to age, type of cancer, marital status, and death in specialised care services (Fig 3c–3f; S4 Table).

Table 3. Utilisation of specialised palliative care services at the end-of-life: overall distribution of cancer deaths from 2013 to 2019 by healthcare region. For each place of death the healthcare region with the lowest proportion of deaths is highlighted in yellow and the highest proportion is highlighted in blue.

| Healthcare region[a] | Home death utilising specialised palliative care services | | Hospital death utilising specialised palliative care services | | Nursing home death utilising specialised palliative care services | | Death in other place [b] | Overall utilisation of specialised palliative care services at death |
|---|---|---|---|---|---|---|---|---|
| | No [b, c] | Yes [b, d] | No [b, e] | Yes [b, f] | No [b, g] | Yes [b, h] | | |
| North region | 1,620 (10.7%) | 1,561 (10.3%) | 5,583 (36.9%) | 1,314 (8.7%) | 3,406 (22.5%) | 899 (5.9%) | 759 (5.0%) | 3,774 (24.9%) |
| Uppsala-Örebro region | 4,540 (13.0%) | 3,976 (11.4%) | 11,588 (33.3%) | 5,610 (16.1%) | 8,544 (24.6%) | 393 (1.1%) | 142 (0.4%) | 9,979 (28.7%) |
| Stockholm region | 666 (2.3%) | 4,280 (15.0%) | 6,488 (22.8%) | 11,934 (41.9%) | 3,354 (11.8%) | 424 (1.5%) | 1,341 (4.7%) | 16,638 (58.4%) |
| West region | 4,410 (16.0%) | 1,599 (5.8%) | 8,973 (32.5%) | 3,055 (11.1%) | 6,586 (23.9%) | 2,663 (9.6%) | 325 (1.2%) | 7,317 (26.5%) |
| Southeast region | 3,038 (17.4%) | 2,448 (14.0%) | 4,799 (27.5%) | 1,711 (9.8%) | 4,836 (27.7%) | 624 (3.6%) | 25 (0.1%) | 4,783 (27.4%) |
| South region | 3,418 (11.8%) | 4,221 (14.6%) | 8,636 (29.9%) | 4,481 (15.5%) | 6,148 (21.3%) | 1,105 (3.8%) | 891 (3.1%) | 9,807 (33.9%) |
| Total population | 17,698 (11.6%) | 18,086 (11.9%) | 46,093 (30.2%) | 28,113 (18.4%) | 32,876 (21.6%) | 6,110 (4.0%) | 3,486 (2.3%) | 52,309 (34.3%) |

Notes

[a]Missing data: Healthcare regions 0.03%.

[b]Row percentages and (n).

[c]General home care.

[d]Specialised palliative home care.

[e]Hospital wards in various medical specialities.

[f]Specialised palliative care wards.

[g]Nursing homes.

[h]Municipality hospices.

For people residing in nursing homes (n = 7,218), no overall longitudinal trend in place of death was detected (0.45, CI: -0.04,0.93) (Fig 2b), and the only covariate with a significant interaction with time was utilising specialised palliative care service at death (Fig 4; S4 Table). However, two regions (Stockholm and West Regions) showed upward trends for dying in hospital among this group (Fig 2). While there was a significantly higher proportion of men than women dying in hospitals (mean difference 4.7%-points), for those residing in nursing homes no interaction was identified between sex and place of death in hospitals (Fig 4a). Upper gastrointestinal and breast and gynaecological cancer showed an upward trend for dying in hospitals, while there were no trends for the other cancer types (Fig 3e). For patients with a palliative care diagnosis and for utilising specialised palliative care services at death, there were also upward trends for dying in hospital (Fig 4b,4f).

## DISCUSSION

### Discussion of Main Findings

The overall time trend identified from 2013–2019 was an increase in home deaths and a decrease in hospital deaths; however, hospital was still the place of death for almost every second person (48.7%) dying from cancer in Sweden, even at the end of this period. Variations across cancer types confirm previous research [11] (e.g., hospital most likely place of death for haematological cancer), while there were cross-regional variations in relation to types of cancer and utilsing specialised palliative care services at death. While a longitudinal overall downward trend was identified for hospital deaths for those residing at home, there were no trend for those residing in nursing homes, with several influencing factors. Taken together, it is thus doubtful the results confirm to any great degree the hypothesis that a national palliative care policy shapes healthcare services in ways that influence place of death for people dying with cancer with a study period of seven years, and especially no influence that

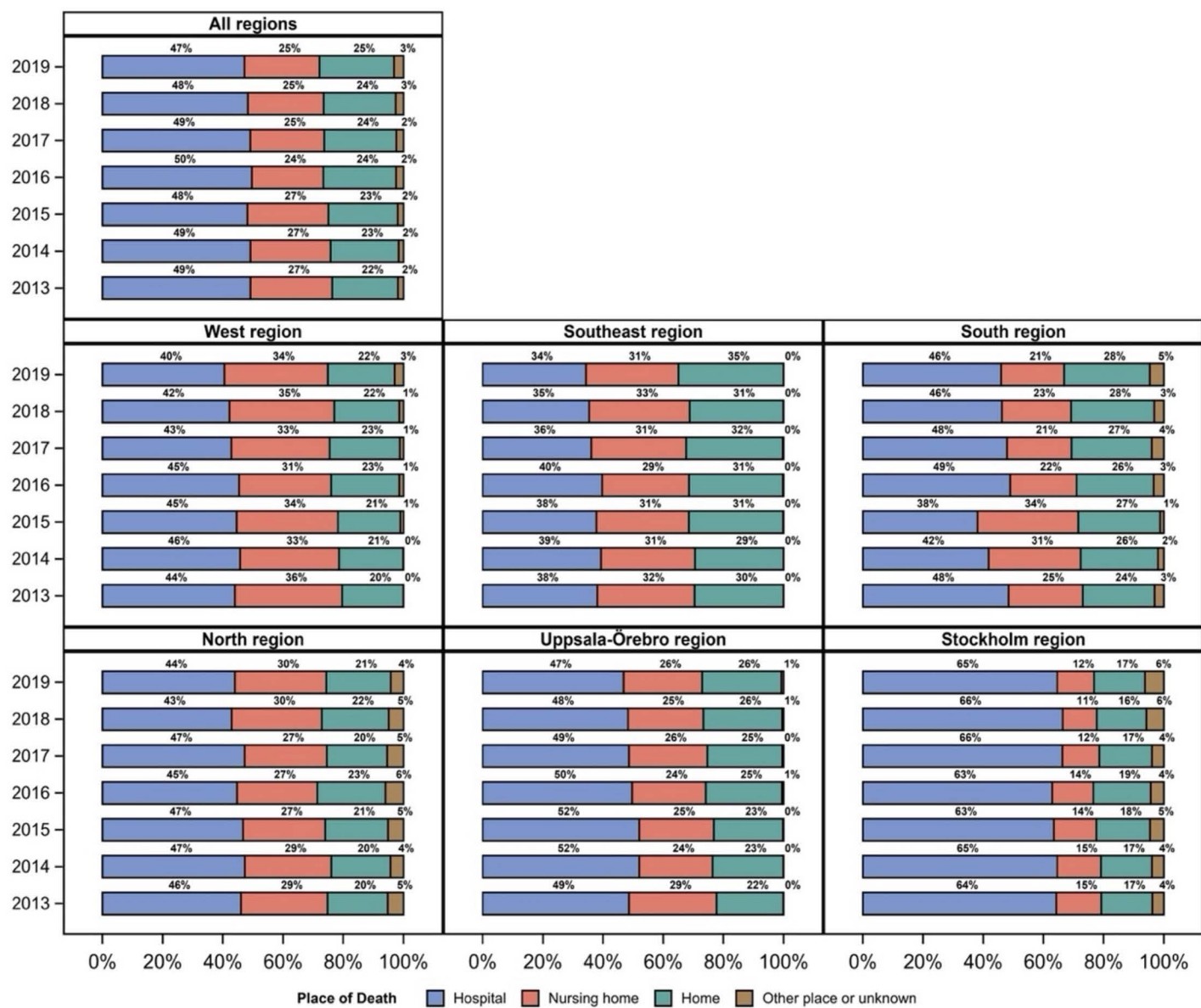

**Fig 1. Distribution of place of death due to cancer in Sweden from 2013 to 2019: total population and within the six healthcare regions.**

aligns with population equity. This result is further reflected by a recent analysis of national disease-specific policy documents in Sweden showing great variation in how palliative care is included. Specifically, there were four different palliative care concepts at play in these policy documents [31].

Nevertheless, the results do confirm a Norwegian study [32] indicating that a national policy may influence place of death, although not fully the patterns of a decrease in hospital deaths and increase in home deaths. The results also confirm that beyond type of cancer and healthcare region, place of death is influenced by socio-demographics. While Norwegian data indicates age, sex and epidemiological changes to be major explanatory factors for changes in place of death [33], our study also shows people who live alone are less likely to die in their own home.

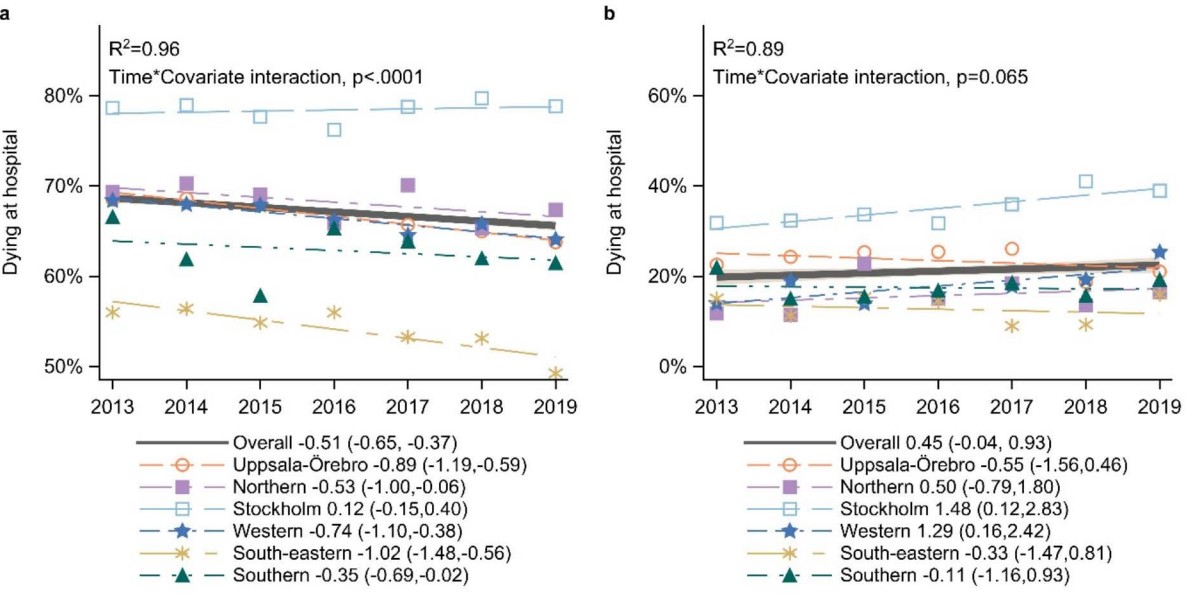

**Fig 2. Longitudinal trends in place of death based on linear regression.** (a) the proportion of people dying in hospital for those residing in their own home (and dying in either hospitals or at home) (n = 103,836), (b) the proportion of people dying in hospital for those residing in nursing homes (and dying in either hospitals or nursing homes) (n = 7,218); each figure showing for the total cancer population (black lines) and the interaction for the overall trend in the total cancer population with healthcare regions (coloured lines). R² is the fraction of the total variation at the population level that may be explained by longitudinal trends across regions.

Considering age, there was no trend identified for those residing at home aged 18 to 59 and a downward trend for dying in hospitals for those 60 years and older, with a stronger trend for the oldest aged 80 and over. Still, one out of five of those 80 years and older died at home, with the remainder almost equally distributed for hospital deaths and nursing home deaths. In an international perspective, the proportion of people dying in nursing homes is high in Sweden. Notably, the proportion of people in the cancer population dying in nursing homes was lower (25.6%) than the general population during the same period (37.7%) [26]. This points to the need for further research into governance of palliative care for older people with cancer.

Our results also indicate that utilising specialised palliative care service at the end of life may increase the prevalence of hospital death. However, the supply of specialised palliative care services varies, as reflected in this study. Regarding palliative care in nursing homes, known differences in local organisation and practice in Sweden [34] can explain the regional differences identified. Our results showing that the percentage of the population with a palliative care diagnosis (30.4%) is smaller than the percentage utilising specialised palliative care services at death (36.3%) is puzzling. We have no explanation for why patients utilising specialised care did not all have a palliative care diagnosis, apart from the fact that the practice of using a palliative care diagnosis is not widespread and varies within the Swedish healthcare services.

Regarding patterns for place of care for the general Swedish population [26], the high proportion of hospital deaths for the total cancer population (48.7%) is distinctive. It contrasts with, for example, cardiovascular disease, which is the most common cause of death but has seen an overall decrease in hospital deaths from 39.1% in 2013 to 26.1% in 2019. Conversely, the proportion of people dying at home for all types of cardiovascular diseases increased from 20.6% in 2023 to 23.4% in 2019 [35]. Given the history of palliative care focusing on the cancer population and challenges to the inclusion of the cardiovascular population, it is notable that the proportion of hospital deaths is clearly highest in the cancer population.

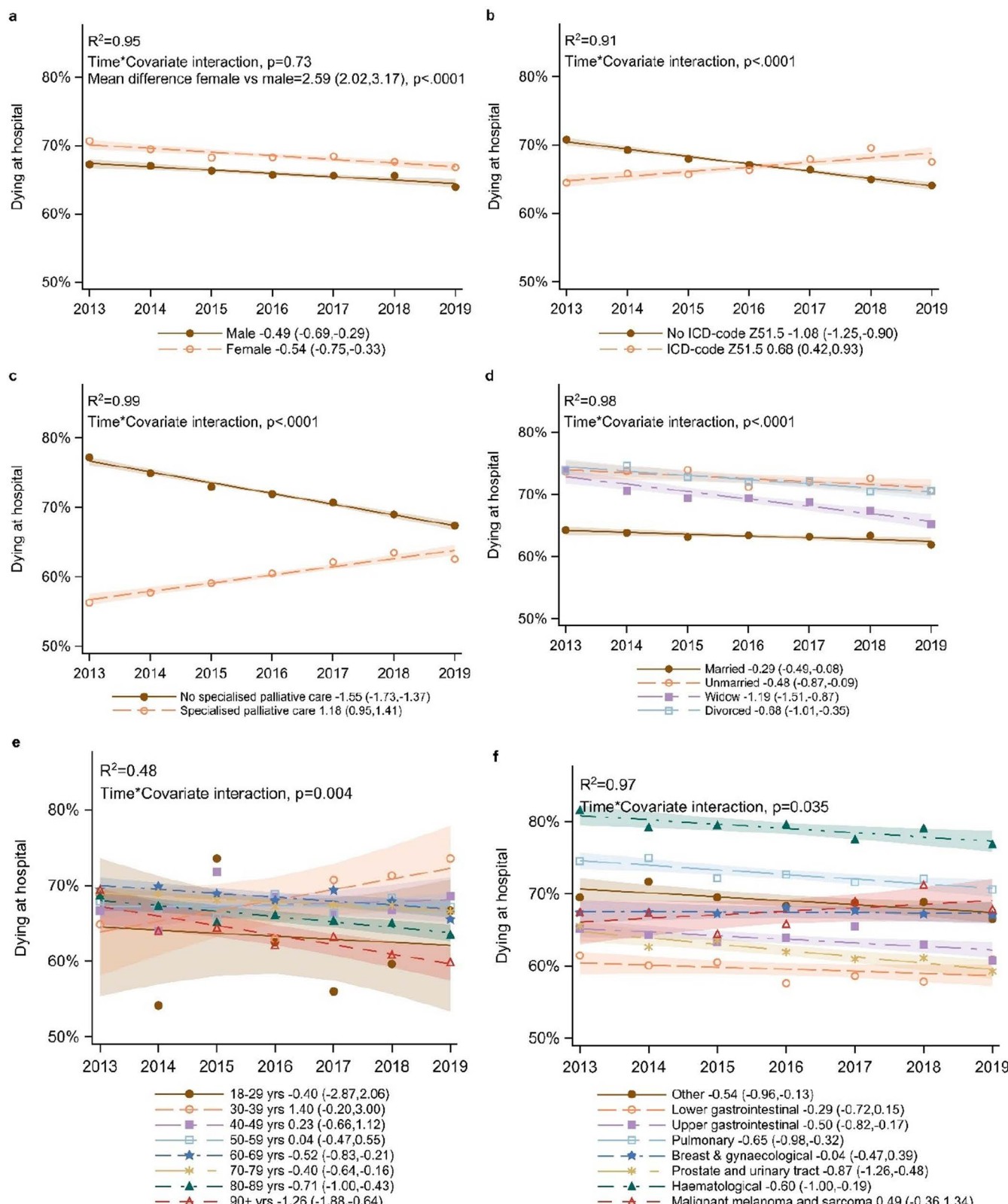

**Fig 3 a-f. Interaction between longitudinal trends in place of death for people residing in own homes aged 18 and older (and dying in either hospitals or at home) and the covariates.** (a) sex, (b) palliative care diagnosis, (c) death in specialised palliative service, (d) marital status, (e) age, (f) cancer type. $R^2$ is the fraction of the total variation at the population level that may be explained by longitudinal trends across subgroup.

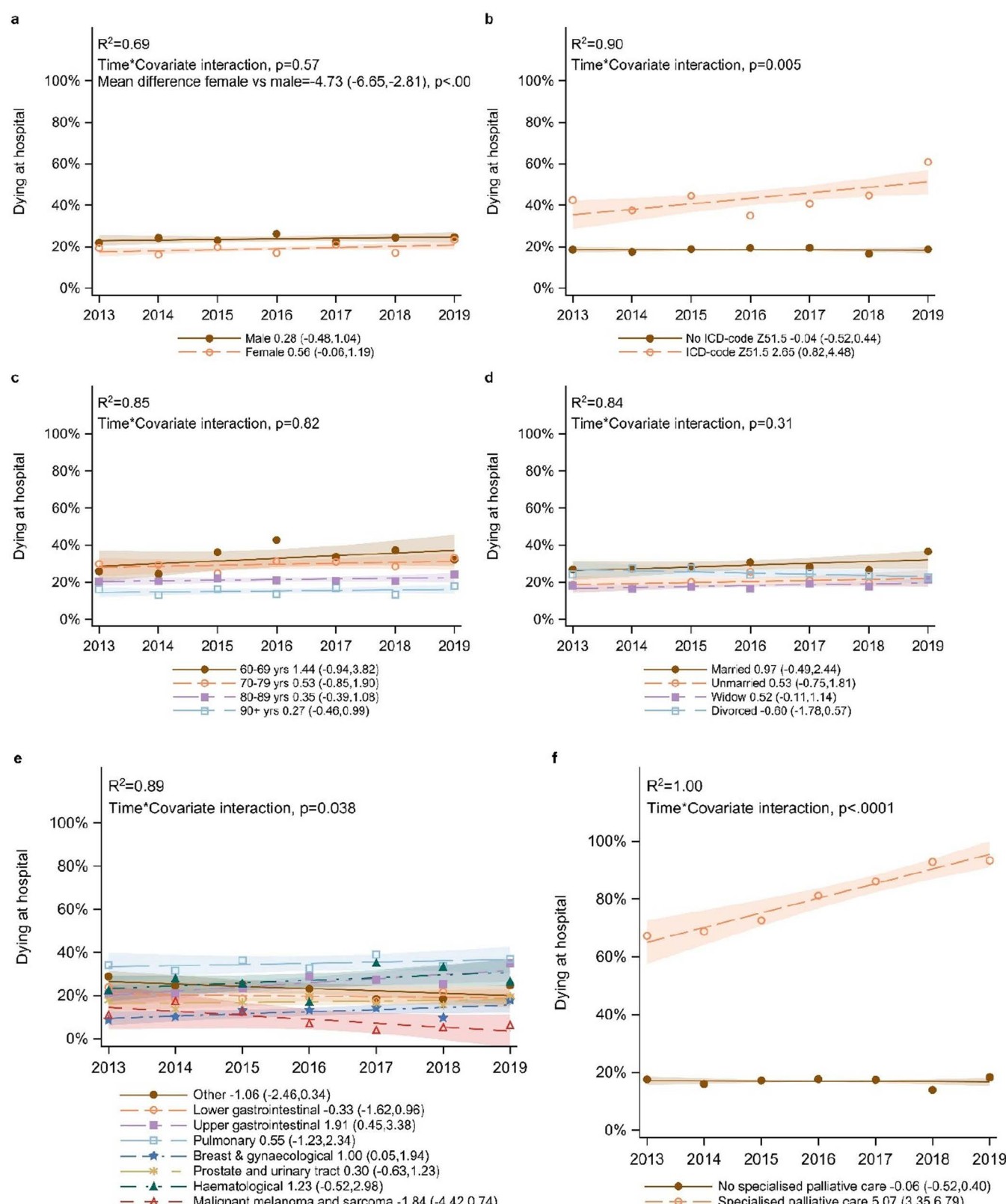

**Fig 4 a–f. Interaction between longitudinal trends in place of death for people residing in nursing homes aged 60 and older (and dying in either hospitals or nursing homes) and the covariates.** (a) sex, (b) palliative care diagnosis, (c) age, (d) marital status, (e) cancer type, and (f) death in specialised palliative services.

Reflecting a weak influence from a national policy for palliative cancer care, the results may also point to a policy-implementation gap related to several factors, including vague adoption of a receptive context for change and environmental pressure, as well as the quality and coherence of the policy in place and how it has been implemented in decision-making [36]. Possible explanations for weak influence on policy have been given by the Lancet Oncology Commission [22], i.e., silo effects at different levels of the healthcare system due to insufficient linkage between healthcare services to target individual patients' varying care needs, as well as the perception of palliative and oncology care as mutually exclusive. The commission's solution for counteracting such factors is an integration of palliative and oncology care early on in patients' cancer trajectories, with coordination of activities for the individual patient and a change at system level. Although early integrated palliative cancer care shows promising results in the Nordic countries [37,38] it is still clearly underutilised and lacking integration of indicators for potential overtreatment [14]. Future national implementation of early integrated palliative cancer care should influence pre-requisites for palliative care reflected in patterns for place of death. This might be especially motivated due to the change in oncological treatments possible for advanced cancer with diverse patient illness trajectories at the end-of-life [39]. In turn, this suggests research into cancer care and treatments in the last year of life further inform the development of integrated palliative cancer care to be initiated earlier on in patients' trajectories.

Possible explanations might also include the influence of national healthcare system infrastructures. Here, by examining various European primary healthcare system models, such as tax-driven ones (i.e., Beveridge models applied in Scandinavia and Britain, for example), social models driven by multiple health insurances (i.e., Bismarck models), and mixed and out-of-pocket models, we may be given an explanatory lens. The Swedish healthcare system aligns to a tax-financed policy driven model [40] Palliat Care albeit with comparatively great regional and municipal freedom in organising services [29]. This is shown in policy at the governmental level providing guidance for governance and organisation; stringent governmental authority in relation to regions and municipalities primarily concerns the provision of health care, with non-binding guidance on *how* to provide it. Our results possibly point to a weak national palliative cancer care policy. When we consider person-centredness as a feature of palliative cancer care, we see that the Swedish population generally rates their healthcare as less person-centred than other populations in Europe [29].

Although home is the most preferred place of death, including for the cancer population [], preferences are also known to be influenced by a range of dimensions in contemporary notions about death and dying [41]. However, most population-based studies on preferences for place of death are based on general populations, and receiving palliative antitumoral treatments might influence and change a patient's preferences. Moreover, undisclosed preferences for place of death have been identified as a major barrier for home deaths, with the patient's family members' preference for the final place of care having a major impact on place of death, while the patient's preference for place of care has a much lower impact. Here, individual studies from cancer care indicate a need for further development of advance care planning [42], changing health professionals' and especially physicians' attitudes towards early integration of palliative care [43], and care planning that incorporates a palliative approach [44] in person-centred ways [45], necessitating the involvement of family members.

## Strengths and limitations

We regard the population-based data and comprehensive data sets linked with a robust analysis as strengths. Thus, the population-level longitudinal analysis provides a high level of observational evidence. However, the major associated limitation here is the inability to

establish causality. Further, the study period, which extends from the year the first national palliative care policy began to be implemented and seven years, may have been too short to identify longitudinal trends in place of death. We have tried to mitigate this through methodological and theoretical considerations. The fact that the years prior to introducing national policy were not included might mask possible trends already in motion at the time of the policy implementation. Still, we assume it is not likely that strong trends for palliative home care were in place in Sweden before 2013.

A major limitation is that some factors already known to influence place of death were not included, such as the patient's preferred place of death, living in a socio-economically deprived area, ethnicity, functional status, intensity of home care use, support from care within civil society, family carer resources, availability and support, integration of home and hospital care services, and to what extent the services had multidisciplinary teams [17]. For place of death in hospitals, no differentiation was made for death in emergency rooms. The major strength of this study is the population design, based on national registers known for high quality data.

Utilising specialised palliative care services at death was based on data from the SRPC because specialised palliative care service is not included as place of death in the Swedish death certificate register. The SRPC national quality register is known to cover close to all deaths in specialised palliative care services in Sweden. The remaining individuals were assumed to not have utilised specialised palliative care services at death, which may have resulted in an overestimation of deaths in this group. However, this means that the population was not limited to the number of deceased people included in the SRPC; the population was defined by people included in the Death certificate register, which has a high coverage.

## Conclusion

This comprehensive analysis sheds light on the complex dynamics influencing place of death for the cancer population, highlighting the limited effect of national palliative care policies and the ongoing challenges in aligning end-of-life care with patient preferences and needs. The results reveal differences in trends based on residence (own home vs. nursing home) with Sweden characterised by a high proportion of hospital deaths and low proportions of home and nursing home deaths, in combination with disparate patterns pertaining to place of death as related to cancer types, age, marital status and healthcare regions, as well as having the palliative care diagnosis and utilising specialised palliative care services at the end-of-life. Thus, the results reflect a need for an impactful, strengthened national palliative care policy that informs governance on the governmental, regional and municipality levels. This probably needs to be combined with further development of effective knowledge translation strategies to drive equitable and person-centred palliative care with impact on governance, organisation and point of care [Cf. 45]. Considering demographic changes with an expected increase in the older population and the observed disparities in care, the need for improved national strategies for palliative cancer care is urgent. With this in mind, it is reasonable to conclude that previous knowledge about inequity in cancer diagnosis, treatment and survival [46] extends to inequities in palliative cancer care.

## Supporting information

**S1 Table. Cross-regional population characteristics**
(PDF)

**S2 Table. Distribution of place of cancer deaths in non-specialised and specialised palliative care services from 2013 to 2019 by healthcare region and cancer types**
(PDF)

**S3 Table. Proportion of i) Home, ii) Hospital, and iii) Nursing home deaths per year and as related to the six healthcare regions**
(PDF)

**S4 Table. Interaction between longitudinal trends in place of death and covariates for a) people residing in own homes aged 18 and older, and for b) people residing in nursing homes aged 60 and older**
(PDF)

## Acknowledgements

We acknowledge everyone who have contributed with data to the registers and the contributions of all the register holders. Thanks to Drs. Henrik Imberg, Nils-Gunnar Pehrsson and Hussein Hamoodi for analytical considerations and statistical analyses, and to members of the Place of death research group for valuable comments and input. This article has been proofread for language purposes by Anita Shenoi.

## Author contributions

**Conceptualization:** Joakim Öhlén, Nyblom Stina, Ozanne Anneli, Nilsson Stefan, Gyllensten Hanna, O'Sullivan Anna, Fürst Carl Johan, Larsdotter Cecilia.

**Data curation:** Joakim Öhlén, Larsdotter Cecilia.

**Formal analysis:** Joakim Öhlén, Nyblom Stina, Ozanne Anneli, Gyllensten Hanna, O'Sullivan Anna, Fürst Carl Johan, Larsdotter Cecilia.

**Funding acquisition:** Joakim Öhlén, Nyblom Stina, Ozanne Anneli, Gyllensten Hanna, Fürst Carl Johan, Larsdotter Cecilia.

**Investigation:** Joakim Öhlén, Nyblom Stina, Ozanne Anneli, Nilsson Stefan, Gyllensten Hanna, O'Sullivan Anna, Fürst Carl Johan, Larsdotter Cecilia.

**Methodology:** Joakim Öhlén, Nyblom Stina, Ozanne Anneli, Gyllensten Hanna, Fürst Carl Johan, Larsdotter Cecilia.

**Project administration:** Joakim Öhlén, Larsdotter Cecilia.

**Resources:** Joakim Öhlén, Larsdotter Cecilia.

**Visualization:** Joakim Öhlén, Larsdotter Cecilia.

**Writing – original draft:** Joakim Öhlén.

**Writing – review & editing:** Joakim Öhlén, Nyblom Stina, Ozanne Anneli, Nilsson Stefan, Gyllensten Hanna, O'Sullivan Anna, Fürst Carl Johan, Larsdotter Cecilia.

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
