## [Decision Letter · Decision Letter 0]

22 Dec 2024

Dear Dr. Öhlén,

We look forward to receiving your revised manuscript.

Kind regards,

Eugenio Paci, MD

Academic Editor

PLOS ONE

Journal Requirements:

“The study was supported by the Swedish state under the agreement between the Swedish government and the county councils the ALF-agreement (ALFGBG-965941); and The Swedish Cancer Society (grant no. 21 1580Pj01H).”

4. In this instance it seems there may be acceptable restrictions in place that prevent the public sharing of your minimal data. However, in line with our goal of ensuring long-term data availability to all interested researchers, PLOS’ Data Policy states that authors cannot be the sole named individuals responsible for ensuring data access (http://journals.plos.org/plosone/s/data-availability#loc-acceptable-data-sharing-methods).

Additional Editor Comments:

This is an interesting paper assessing the impact of palliative care policy on the place of death in a large country. Results on the same data analyzing trends in the same period were already published (ref 26). In this manuscript, the authors referred to cancer types. This might be an interesting sub-analysis of the data. The presentation of the data follows the previous pattern, as in the tables and statistical methodology. An effort to compare the total and cancer-specific cancers might probably confirm with this analysis (and some information in the paper about cardiovascular conditions) that major determinants are contextual (area, age groups, SES, etc) more than attributable to specific pathology (with exception also from literature for hematological neoplasms).

The trend considered in this paper is 2013-2019, after the 2012 recommendations and innovations in policy. There is no reference to the possible existence of a trend before 2012. For example, in a study in Germany from 2001 to 2011, there was a decreasing trend in this period; certainly, the issue of hospital deaths was at the center of interest before 2012, and a decreasing trend might have started after 2000. Sweden has a high proportion of deaths in nursing homes and the proportion is stable and high compared with other countries. So, this may explain the slow change in-hospital deaths after 2012.

The study investigated how the introduction of palliative care diagnosis (it needs a better explanation, as requested by the reviewer) in different settings (in-hospital palliative care) might reduce the need to transfer the cancer patient to home. At leat in my knowledge palliative care was mainly at home.

Finally, possibly a more in-depth knowledge of the last year of life is important. In recent years, the treatment of metastatic cancer has changed and possibly is much more medicalized than before. Perhaps, a better knowledge of the end-of-life pattern of oncological care should be relevant in explaining the place of death.

Reviewers' comments:

Reviewer's Responses to Questions

**Comments to the Author**

1. Is the manuscript technically sound, and do the data support the conclusions?

Reviewer #1: Partly

2. Has the statistical analysis been performed appropriately and rigorously?

Reviewer #1: Yes

3. Have the authors made all data underlying the findings in their manuscript fully available?

Reviewer #1: Yes

4. Is the manuscript presented in an intelligible fashion and written in standard English?

Reviewer #1: Yes

Reviewer #1: I thank the Academic Editor for giving me the opportunity to read the manuscript “Impact of a national palliative care policy on place of death for people with different cancer types: a national population-based study in a high-income country with an increasing proportion of older people” by Joakim Öhlén and Colleagues (PONE-D-24-38475). There follow some suggestions that I suppose may be helpful to improve the manuscript. I have numbered the pages starting with 1 on the title page.

Title

I suggest the authors to consider the possibility to modify it. The term ‘impact’ seems inappropriate, because the authors do not have historical reference data. Also, the country of origin of the paper needs to be mentioned. Conversely, I think that the notation “ ... in a high-income country with an increasing proportion of older people” is rather unnecessary. Besides, patient age is not the only factor being investigated in the study.

Page 6. “ … living in rural areas is associated with hospital deaths [13]”.

Please note that, in the next sentence, an opposite association is mentioned (rural areas and home deaths). I suppose that there are inconsistencies between the results of cited studies.

Page 7. Last sentence of the top paragraph.

This is a strong and important statement. Does it have some relationship with the next one (see below)?

Page 7. Bottom paragraph.

The statement “ … the hypothesis is that end-of-life and palliative care policy shape healthcare services, which in turn influence service utilization and ultimately place of death” seems to be central to the rationale of the study but is too concise and somewhat unclear. In particular, it is unclear to me what “ … end-of-life and palliative care policy shape healthcare services …” means. The study hypothesis needs to be explained better. The same statement is in the Abstract too. Perhaps a brief clarification of terms can be added.

Page 8. Lines 3-6.

“… recent place of death result for the total population”. Do the authors mean “… results of a recent study on the place of death for the total population”?

Page 8. Bottom line.

Please include ‘nationwide’ in the study definition. Also, although two references are provided to the reader, a very concise description of the national guidelines of 2012 would be useful.

Page 9. Top paragraph.

Record linkage procedures are an important methodological issue. Have the authors used a personal universal or unique identifying number? If so, I do not understand the need for anonymising the records. Some more detail of the health record linkage procedures that are available in Sweden would be of interest to the reader.

Page 10. Line 3.

Perhaps Beverage should be changed to Beveridge.

Page 11. Study variables.

The place of death was classified into “hospital, home, nursing home, other”. In the footnote to Table 1 (page 15), it reads that specialised palliative services included specialised palliative home care, specialised palliative hospital care, and municipality hospice care. This seems to suggest that there could also be non-specialised palliative home care services and non-specialised palliative hospital care services. Can the authors provide some details or examples? Secondly, in which category of the four-tier classification are municipal hospices classified? The nomenclature used should be clarified and completed in the body of the text (the reader would notice the footnote to Table 1 too late …). Finally, please clarify “… except from other places …”.

Page 17 (and elsewhere). “… palliative care diagnosis …”.

Unfortunately, I do not know the meaning of this term. Please provide a definition.

Table 3.

In Table 3, the columns headed ‘No’ are very difficult to interpret. The sum of the row percentages is 100%. Apparently, this means that –for example– a person who did not die at home (first column) could not die elsewhere. Of course, this cannot be the case. Please clarify.

Page 28. Bottom lines.

The age groups 80-89 years and 90+ showed a downward trend to die in hospital. Was this decrease associated with an increasing trend to die at home? This is probably shown in the Figure, but the resolution in the PDF is not good. If an increasing trend to die at home was observed in the elderly, this (unexpected, I guess) finding would need a comment.

Page 31. Lines 1-4.

Please explain more explicitly why the study hypothesis was not confirmed.

Page 32. Line 6.

“… decreased from 20.6% in 1023 to 23.4% in 2019…”. In fact, this is an increase.

Page 33. Middle paragraph.

“… comparatively great regional and municipal freedom in organising services …”. I suggest the Authors to discuss more deeply this key issue.

Page 34. Strengths and limitations.

I agree that the exclusion of the years prior to the introduction of the new Guidelines is a limitation of the study. However, this does not mean that a sound conclusion cannot be drawn from the data. The suboptimal situation observed, indeed, points to the need for an improvement. From this point of view, the problem of the relatively limited time span of the study is mitigated by a consideration: this study was undertaken in a timely manner. In addition, the results are disappointing enough to represent a baseline from which to improve.

Page 36. “… the need for improved national strategies for palliative cancer care …” and “... a great need for national strategies for governance …”.

An improvement of palliative care for cancer is undoubtedly needed, but I suggest the authors to centre their conclusions on the issue of the effectiveness of Guidelines of 2012, in line with the rationale of the study. For example, the results might be interpreted to show (1) that a revision of the national policy is needed or (2) that there is only a problem of insufficient implementation or (3) that it is premature to make a decision about the policy and that other years of monitoring are necessary – and so on. Also, assuming that the authors support a revision, which priorities does the study suggest (if any)?

**Do you want your identity to be public for this peer review?** For information about this choice, including consent withdrawal, please see our Privacy Policy

Reviewer #1: No

---

## [Author Response · Author response to Decision Letter 1]

26 Jan 2025

We have described editor and reviewer comments in the new cover letter and in the rebuttal file.

---

## [Decision Letter · Decision Letter 1]

13 Feb 2025

Influence of palliative care policy on place of death for people with different cancer types: a nationwide register study

PONE-D-24-38475R1

Dear Dr. Öhlén,

We’re pleased to inform you that your manuscript has been judged scientifically suitable for publication and will be formally accepted for publication once it meets all outstanding technical requirements.

Kind regards,

Eugenio Paci, MD

Academic Editor

PLOS ONE

Additional Editor Comments (optional):

Reviewers' comments:

Reviewer's Responses to Questions

**Comments to the Author**

Reviewer #1: All comments have been addressed

2. Is the manuscript technically sound, and do the data support the conclusions?

Reviewer #1: Yes

3. Has the statistical analysis been performed appropriately and rigorously?

Reviewer #1: Yes

4. Have the authors made all data underlying the findings in their manuscript fully available?

Reviewer #1: Yes

5. Is the manuscript presented in an intelligible fashion and written in standard English?

Reviewer #1: Yes

Reviewer #1: I thank the Authors for the changes made in this second version of the manuscript and for the clarifications provided.

**Do you want your identity to be public for this peer review?** For information about this choice, including consent withdrawal, please see our Privacy Policy

Reviewer #1: No

---

## [Editor Report · Acceptance letter]

PONE-D-24-38475R1

PLOS ONE

Dear Dr. Öhlén,

I'm pleased to inform you that your manuscript has been deemed suitable for publication in PLOS ONE. Congratulations! Your manuscript is now being handed over to our production team.

Kind regards,

on behalf of

Dr. Eugenio Paci

Academic Editor

PLOS ONE